# IMPROVED CONVERGENCE RATE FOR DIFFUSION PROBABILISTIC MODELS

**Gen Li** *
Department of Statistics
The Chinese University of Hong Kong
Hong Kong
genli@cuhk.edu.hk

**Yuchen Jiao***
Department of Statistics
The Chinese University of Hong Kong
Hong Kong
yuchenjiao@cuhk.edu.hk

## ABSTRACT

Score-based diffusion models have achieved remarkable empirical performance in the field of machine learning and artificial intelligence for their ability to generate high-quality new data instances from complex distributions. Improving our understanding of diffusion models, including mainly convergence analysis for such models, has attracted a lot of interests. Despite a lot of theoretical attempts, there still exists significant gap between theory and practice. Towards to close this gap, we establish an iteration complexity at the order of $d^{1/3}\varepsilon^{-2/3}$, which is better than $d^{5/12}\varepsilon^{-1}$, the best known complexity achieved before our work. This convergence analysis is based on a randomized midpoint method, which is first proposed for log-concave sampling (Shen & Lee, 2019), and then extended to diffusion models by Gupta et al. (2024). Our theory accommodates $\varepsilon$-accurate score estimates, and does not require log-concavity on the target distribution. Moreover, the algorithm can also be parallelized to run in only $O(\log^2(d/\varepsilon))$ parallel rounds in a similar way to prior works.

## 1 INTRODUCTION

Score-based diffusion models are a class of generative models that have gained prominence in the field of machine learning and artificial intelligence for their ability to generate high-quality new data instances from complex distributions (Dhariwal & Nichol, 2021; Ho et al., 2020; Sohl-Dickstein et al., 2015; Song & Ermon, 2019; Song et al., 2021). These models operate by gradually transforming noise into samples from the target distribution through a denoising process guided by pretrained neural networks that approximate the score functions. In practice, score-based diffusion models have demonstrated remarkable performance in generating realistic and diverse content across various domains (Croitoru et al., 2023; Ramesh et al., 2022; Rombach et al., 2022; Saharia et al., 2022; Li & Jiao, 2025), achieving state-of-the-art performance in generative AI.

The development of score-based diffusion models is closely related to the theory of stochastic processes. At a high level, we consider a forward process:

$$X_0 \overset{\text{add noise}}{\to} X_1 \overset{\text{add noise}}{\to} \cdots \overset{\text{add noise}}{\to} X_T,$$

which draws a sample from the target data distribution (i.e., $X_0 \sim p_{\mathsf{data}}$), and then progressively diffuses it to Gaussian noise over time. The key step of the diffusion model is to construct a reverse process:

$$Y_T \overset{\text{denoise}}{\to} Y_{T-1} \overset{\text{denoise}}{\to} \cdots \overset{\text{denoise}}{\to} Y_0,$$

satisfying $Y_t \overset{d}{\approx} X_t$ for all $t$, which starts with pure Gaussian noise (i.e., $Y_T \sim \mathcal{N}(0, I_d)$) and gradually converts it back to a new sample $Y_0$ sharing a similar distribution to $X_0$. Evidently, the most crucial step of the diffusion model lies in effective design of the reverse process. To accomplish this goal, $Y_{t-1}$ in each step is typically obtained from $Y_t$ with the aid of score functions

---

*The authors contributed equally.

$(s_t^\star = \nabla \log p_{X_t})$, which are pre-trained by means of score matching techniques (e.g., Hyvärinen & Dayan (2005); Ho et al. (2020); Hyvärinen (2007); Vincent (2011); Song & Ermon (2019); Pang et al. (2020)).

Due to the impressive empirical success, in the past few years, there have been a lot of works exploring the convergence of diffusion models (De Bortoli, 2022; Gao et al., 2023; Lee et al., 2022; 2023; Chen et al., 2022; Benton et al., 2023; Chen et al., 2023; Li et al., 2023; 2024c; Gupta et al., 2024; Chen et al., 2024b; Li et al., 2024a; Li & Yan, 2024a; Huang et al., 2024a; Li & Cai, 2024; Li & Yan, 2024b; Huang et al., 2024b; Liang et al., 2025; Li et al., 2024b). They typically treat the score matching stage as a blackbox and study how the sampling steps $T$ and the score estimation error will affect the sampling accuracy. Given the observation that there still exists a huge gap between prior theories and practice, this paper is devoted to improving the convergence theory for diffusion models.

## 1.1 MOTIVATION

Following the most existing works, let's focus on the total variation distance between distributions of the target data and generated samples. Let $d$ be the dimension, and $\varepsilon$ denote the accuracy which means the total variation is smaller than $\varepsilon$. For the general target distribution, Benton et al. (2023) provides first linear $d$-dependency rate with an iteration complexity $d/\varepsilon^2$. Later on, Li et al. (2024c) achieves an improved complexity scaling as $d/\varepsilon$, while with a burn-in cost $d^2$.

However, such kind of result is still far away from practical observations. For example, for two typical image datasets, CIFAR-10 with $d = 32 \times 32 \times 3 = 3072$ and ImageNet $256 \times 256$ with $d = 256 \times 256 \times 3 = 196608$, it is sufficient to use about 50 steps to generate good samples (Song et al., 2021; Nichol & Dhariwal, 2021), which is much smaller than the theoretical requirement. To close this gap, people try to derive faster convergence rate through exploiting extra data distribution properties, such as low-dimensional structure (Li & Yan, 2024a) and smoothness of score functions (Gupta et al., 2024; Chen et al., 2024b). Following this line of work, the scope of this paper lies in establishing some convergence theory for diffusion models towards aligning more closely with practical requirements.

## 1.2 OUR CONTRIBUTIONS

Focusing on the diffusion models with $L$-smooth score functions, this paper develops an improved iteration complexity as following (up to some logarithmic factors):

$$Ld^{1/3}\varepsilon^{-2/3},$$

which is inspired by the randomized mid-point design (Shen & Lee, 2019; Gupta et al., 2024). In what follows, let us make a brief comparison between our results and the state-of-the-art results about the convergence rate of score-based generative models.

- *Comparison under smoothness condition.* Similarly, Chen et al. (2024b) also studied the convergence theory by assuming smooth score functions. Recently, the iteration complexity is improved by Gupta et al. (2024); Chen et al. (2024b) in this setting, which, to ensure $\varepsilon$-accuracy, scales as $\tilde{O}(L^{5/3}d^{5/12}\varepsilon^{-1})$. This means that our convergence rate improves prior works by a factor of $L^{2/3}d^{1/12}\varepsilon^{-1/3}$.

- *Comparison under general condition.* Without the smoothness condition on score functions, Benton et al. (2023) proved an iteration complexity at the order of $d\varepsilon^{-2}$, which first exhibits linear dependency in the data dimension $d$. This bound is improved to $\widetilde{O}(d/\varepsilon)$ by Li et al. (2024c) when the number of steps is larger than $\widetilde{O}(d^2)$. In comparison, to achieve $\varepsilon$-accuracy, our theory improves these works provided that the Lipschitz conditions of score functions satisfy

$$L < d^{2/3}\varepsilon^{-4/3}, \quad \text{for } \varepsilon > d^{-1/2} \qquad \text{or} \qquad L < d^{2/3}\varepsilon^{-1/3}, \quad \text{for } \varepsilon < d^{-1/2}.$$

Hopefully, such relations can be satisfied in most real cases, for example, considering ImageNet $256 \times 256$, this requirement becomes $L < 3381$ for $\varepsilon \approx 1$ and $L < 72845$ for $\varepsilon \approx 0.1$.

## 2 PRELIMINARY

In this section, we introduce some basics and notations of score-based generative models. The diffusion generative model typically encompasses two processes: a forward process and a reverse process, as described below.

**Forward process.** Starting from $X_0$ drawn from some target distribution $p_{\text{data}}$, the forward process evolves as follows:

$$X_t = \sqrt{\alpha_t} X_{t-1} + \sqrt{1 - \alpha_t} W_t, \quad 1 \leq t \leq T, \tag{1}$$

where $0 < \alpha_t < 1$, and $\{W_t\}_{1 \leq t \leq T}$ is a sequence of independent noise vectors drawn from $W_t \overset{\text{i.i.d.}}{\sim} \mathcal{N}(0, I_d)$. In addition, let's define

$$\overline{\alpha}_t := \prod_{k=1}^{t} \alpha_k, \quad 1 \leq t \leq T, \tag{2}$$

which will be helpful in our following analysis. Using this definition, $X_t$ can be expressed in closed form as $X_t = \sqrt{\overline{\alpha}_t} X_0 + \sqrt{1 - \overline{\alpha}_t} \, \overline{W}_t$ with $\overline{W}_t \sim \mathcal{N}(0, I_d)$. The diffusion models draw a lot of motivations from the stochastic differential equations. Hence, let's also introduce the continuous-time limit of the forward diffusion process, which can be modeled as

$$\mathrm{d}X_\tau = -\frac{1}{2(1 - \tau)} X_\tau \mathrm{d}\tau + \frac{1}{\sqrt{1 - \tau}} \mathrm{d}B_\tau, \quad \text{for } 0 \leq \tau < 1, \tag{3}$$

where $B_\tau$ denotes some Brownian motion.

**Reverse process.** The core of diffusion models lies in constructing a reverse-time process with nearly identical marginals as the forward process, namely, $Y_\tau \overset{d}{\approx} X_\tau$. Fortunately, it is shown that there exists some probability flow ODE

$$\mathrm{d}Y_\tau = -\frac{1}{2(1 - \tau)} \big(Y_\tau + \nabla \log p_{X_\tau}(Y)\big) \mathrm{d}\tau, \tag{4}$$

which can achieve our goal, such that if we start from $Y_{\tau_0} \sim p_{X_{\tau_0}}$ for any $0 \leq \tau_0 < 1$, it can be ensured that $Y_\tau \overset{d}{\approx} X_\tau$ for all $0 \leq \tau < 1$. Then the design of reverse process is equivalent to discretize the above ODE process (4), as well as estimate $\nabla \log p_{X_\tau}$, which is also called score function.

**Score functions.** The score functions are crucial for score-based generative modeling, and the definition of score function is as follows.

**Definition 1.** The score function, denoted by $s_t^\star : \mathbb{R}^d \to \mathbb{R}^d (1 \leq t \leq T)$, is defined as

$$s_t^\star(x) := \nabla \log p_{X_t}(x) = -\frac{1}{1 - \overline{\alpha}_t} \int_{x_0} p_{X_0 \mid X_t}(x_0 \mid x)(x - \sqrt{\overline{\alpha}_t} x_0) \mathrm{d}x_0, \quad 1 \leq t \leq T. \tag{5}$$

Following the same way as prior analysis, we assume access to faithful estimates of the score functions $s_t^\star$ across all intermediate steps $t$ as following.

**Assumption 1.** *We assume access to an estimate $s_t(\cdot)$ for each $s_t^\star(\cdot)$ with the averaged $\ell_2$ score estimation error as*

$$\varepsilon_{\text{score}}^2 = \frac{1}{T} \sum_{t=1}^{T} \mathbb{E}_{x \sim q_t} \big[\|s_t(x) - s_t^\star(x)\|_2^2\big] =: \frac{1}{T} \sum_{t=1}^{T} \varepsilon_t^2. \tag{6}$$

In addition, we mainly focus on the case with smooth score functions in this paper, which is stated below.

**Assumption 2.** *Assume that $s_t^\star(x)$ and $s_t(x)$ are Lipschitz for all $t$ such that*

$$\|s_t^\star(x_1) - s_t^\star(x_2)\|_2 \leq L\|x_1 - x_2\|_2; \tag{7a}$$

$$\|s_t(x_1) - s_t(x_2)\|_2 \leq L\|x_1 - x_2\|_2. \tag{7b}$$

**Target data distribution.** Following prior works, we also need the following assumption on the target data distribution $p_{\text{data}}$ to establish our theory.

**Assumption 3.** *We assume that the target distribution $p_{\text{data}}$ has bounded second-order moment in the sense that*

$$R := \mathbb{E}[\|X_0\|_2^2] < T^{c_R} \tag{8}$$

*for arbitrarily large constant $c_R > 0$.*

Here, this assumption requires the second-order moment of $p_{\text{data}}$ to be exceedingly large (given that the exponent $c_R$ can be arbitrarily large), which is a mild assumption. However, there exist exceptions including extremely heavy-tailed or non-smooth distributions. For example, densities decaying slower than $1/x^3$ or purely discrete data may not align with our theoretical framework.

## 3 MAIN RESULTS

In this section, we first introduce a new score-based sampler with randomized midpoints, which is modified from Gupta et al. (2024) for ease of analysis. Then, we prove a faster convergence rate for this sampler based on a new analysis framework, which might be useful for improving convergence guarantees for other variations of samplers. Finally, we also provide a parallel sampling strategy for our sampler following the similar idea from prior works (Gupta et al., 2024; Chen et al., 2024a).

### 3.1 ALGORITHM

This part is devoted to explaining the details of our score-based sampler with randomized midpoints. Before proceeding, let's first specify the choice of learning rates to be used in our sampler.

**Randomized schedule.** Similar to prior works, we adopt the following randomized learning rate schedule

$$\overline{\alpha}_t \sim \mathsf{Unif}(\widehat{\alpha}_t, \widehat{\alpha}_{t-1}), \quad \text{for } t = -\frac{N}{2} + 1, \dots, T + 1, \tag{9a}$$

where Unif denotes the uniform distribution, and

$$\widehat{\alpha}_{T+1} = \frac{1}{T^{c_0}}, \qquad \text{and} \qquad \widehat{\alpha}_{t-1} = \widehat{\alpha}_t + \frac{c_1 \widehat{\alpha}_t (1 - \widehat{\alpha}_t) \log T}{T} \tag{9b}$$

for some sufficiently large constants $c_0, c_1 > 0$ obeying $c_1/c_0$ sufficiently large.

Moreover, for ease of presentation, we let $K = c_2 L \log T$ for some constant $c_2 > 0$ such that $\frac{c_2}{c_1}$ sufficiently large, and $N = \frac{2T}{K}$. Then define for $n = 0, \dots, N$, and $k = 0, \dots, K-1$,

$$\widehat{\tau}_{k,n} := 1 - \widehat{\alpha}_{T - \frac{kN}{2} - n} \qquad \text{and} \qquad \tau_{k,n} := 1 - \overline{\alpha}_{T - \frac{kN}{2} - n + 1}, \tag{9c}$$

which satisfy the following lemma. The detailed proof of this result can be found in Li & Jiao (2024)(Appendix B.1).

**Lemma 1.** *Our choice of learning schedules (9) satisfies*

$$1 - \tau_{0,0} \le \widehat{\alpha}_T \le \frac{2}{T^{c_0}}, \qquad \tau_{K,0} \le 1 - \widehat{\alpha}_1 \le \frac{1}{T^{c_0}}, \qquad \text{and} \qquad \frac{\widehat{\tau}_{k,n-1} - \widehat{\tau}_{k,n}}{\widehat{\tau}_{k,n-1}(1 - \widehat{\tau}_{k,n-1})} = \frac{c_1 \log T}{T}. \tag{10}$$

**Sampling procedure.** With the learning schedule in hand, we are now ready to introduce the sampling procedure. The algorithm is actually a discretization of the probability ODE flow incorporated with some stochastic noise, which proceeds as follows. We start from $Y_0 \sim \mathcal{N}(0, I_d)$, and then for $k = 0, \dots, K-1$, we keep updating $Y_k$ through the formula:

$$Y_{k+1} = \sqrt{\frac{1 - \tau_{k+1,0}}{1 - \tau_{k,N}}} Y_{k,N} + \sqrt{\frac{\tau_{k+1,0} - \tau_{k,N}}{1 - \tau_{k,N}}} Z_k, \tag{11a}$$

where $Z_k \overset{\text{i.i.d.}}{\sim} \mathcal{N}(0, I)$ and for $n = 1, \dots, N$, we compute

$$\frac{Y_{k,n}}{\sqrt{1 - \tau_{k,n}}} = \frac{Y_k}{\sqrt{1 - \tau_{k,0}}} + \frac{s_{T - \frac{kN}{2} + 1}(Y_k)}{2(1 - \tau_{k,0})^{3/2}}(\tau_{k,0} - \widehat{\tau}_{k,0}) + \sum_{i=1}^{n-1} \frac{s_{T - \frac{kN}{2} - i + 1}(Y_{k,i})}{2(1 - \tau_{k,i})^{3/2}}(\widehat{\tau}_{k,i-1} - \widehat{\tau}_{k,i})$$

$$+ \frac{s_{T - \frac{kN}{2} - n + 2}(Y_{k,n-1})}{2(1 - \tau_{k,n-1})^{3/2}}(\widehat{\tau}_{k,n-1} - \tau_{k,n}). \tag{11b}$$

Here, we let $Y_{k,0} = Y_k$. Notice that for each step to calculate $Y_{k,n}$, only one additional score evaluation for $s_{T - \frac{kN}{2} - n + 2}(Y_{k,n-1})$ is needed. This implies that the total iteration complexity is $KN = 2T$.

## 3.2 CONVERGENCE ANALYSIS

We are now positioned to present the convergence guarantees — measured by the total variation distance between the forward and the reverse processes — for the proposed sampler (11). The proof is postponed to Section 4.

**Theorem 1.** *Suppose that Assumptions 1, 2, and 3 hold true. Then the sampling process (11) with the learning rate schedule (9) satisfies*

$$\mathsf{TV}\left(p_{\mathsf{data}}, p_{Y_K}\right) \leq \frac{C L^{3/2} d^{1/2} \log^{5/2} T}{T^{3/2}} + C \varepsilon_{\mathsf{score}} \log^{1/2} T \tag{12}$$

*for some constant $C > 0$ large enough.*

We now take a moment to discuss the main implications of Theorem 1.

**Iteration complexity.** Let's focus on the first term in (12), which corresponds to discretization error, and ignore the score estimation error. To ensure $\mathsf{TV}\left(p_{\mathsf{data}}, p_{Y_K}\right) \leq \varepsilon$, it is sufficient to choose

$$T \gtrsim \frac{L d^{1/3} \log^{5/3} T}{\varepsilon^{2/3}}.$$

As far as we know, the existing iteration complexity guarantees (with sub-linear dependency on $d$) are $\widetilde{O}(L^{5/3} d^{5/12} \varepsilon^{-1})$ achieved in Gupta et al. (2024) and $\widetilde{O}(L^2 d^{1/2} \varepsilon^{-1})$ derived in Chen et al. (2024b). Our theory improves the prior results with at least $\widetilde{O}(L^{2/3} d^{1/12} \varepsilon^{-1/3})$.

**Dependency of score estimation error.** Turning attention to the second term, the sampling accuracy scales as $\varepsilon_{\mathsf{score}} \log^{1/2} T$, suggesting that our sampler is stable to imperfect score estimation. In comparison, the existing results (with sub-linear dependency on $d$) scale as $L^{1/2} d^{1/12} \varepsilon_{\mathsf{score}} \log T$ and $L^{1/2} \varepsilon_{\mathsf{score}} \log T$ in Gupta et al. (2024) and Chen et al. (2024b), respectively. Our theory exhibits a better dependency on score estimation error, which is similar to the results in Chen et al. (2022) and Benton et al. (2023).

**Lipschitz condition of score functions.** It is noteworthy that our better iteration complexity holds under the condition of Lipschitz continuity for score functions. Without this condition, the best results are $\widetilde{O}(d\varepsilon^{-2})$ and $\widetilde{O}(d\varepsilon^{-1} + d^2)$, which are achieved in Benton et al. (2023) and Li et al. (2024c), respectively. Our theory is better than such result when the Lipschitz constant for the score functions satisfies $L < d^{2/3} \varepsilon^{-4/3}$ for $\varepsilon > d^{-1/2}$, or $L < d^{2/3} \varepsilon^{-1/3}$ for $\varepsilon < d^{-1/2}$, and it is still unknown how to achieve sub-linear dependency on $d$ in the general case.

Finally, we remark that the extension to non-Lipschitz settings is significantly more challenging due to the critical role the smoothness assumption plays in our analysis. Previous analysis on diffusion models focus primarily on stepwise error propagation, analyzing how errors at one step affect the next. In contrast, our randomized design requires tracking error propagation across multiple steps simultaneously, which necessitates uniform control over these errors. Without smoothness, only a high-probability bound that depends on the dimension $d$ can be established, which is hard to use compared to the uniform bound and may result in worse bound since it scales with $d$.

### 3.3 PARALLEL SAMPLING

In this section, we provide a way to do parallel sampling for our proposed sampler that needs the similar number of parallel rounds as prior works (Gupta et al., 2024), while the number of parallel processors here is significantly reduced. In the following, we first introduce the parallel sampling procedure, and then present the theoretical guarantee. The analysis is postponed to the Section 5.

**Details of parallel sampling procedure.** In the parallel sampling, all procedures are the same as the above sampler (11), except that we will update $Y_{k,n}$ for $n = 1, \cdots, N$ at the same time. Specifically, we first initialize

$$\frac{Y_{0,k,n}}{\sqrt{1-\tau_{k,n}}} = \frac{Y_k}{\sqrt{1-\tau_{k,0}}}, \quad n = 1, \cdots, N. \tag{13a}$$

Then we update all $Y_{k,n}$ for $M$ rounds simultaneously: at the $m$-th round, we update $Y_{m,k,n}$ for $n = 1, \ldots, N$ in the following way:

$$\frac{Y_{m,k,n}}{\sqrt{1-\tau_{k,n}}} = \frac{Y_k}{\sqrt{1-\tau_{k,0}}} + \frac{s_{T-\frac{kN}{2}+1}(Y_k)}{2(1-\tau_{k,0})^{3/2}}(\tau_{k,0} - \widehat{\tau}_{k,0}) + \sum_{i=1}^{n-1} \frac{s_{T-\frac{kN}{2}-i+1}(Y_{m-1,k,i})}{2(1-\tau_{k,i})^{3/2}}(\widehat{\tau}_{k,i-1} - \widehat{\tau}_{k,i})$$

$$+ \frac{s_{T-\frac{kN}{2}-n+2}(Y_{m-1,k,n-1})}{2(1-\tau_{k,n-1})^{3/2}}(\widehat{\tau}_{k,n-1} - \tau_{k,n}). \tag{13b}$$

Finally, we compute

$$Y_{k+1} = \sqrt{\frac{1-\tau_{k+1,0}}{1-\tau_{k,N}}} Y_{M,k,N} + \sqrt{\frac{\tau_{k+1,0}-\tau_{k,N}}{1-\tau_{k,N}}} Z_k, \tag{13c}$$

and we complete the update from $Y_k$ to $Y_{k+1}$ in the parallel way. In this parallel setting, the total number of parallel rounds to yield sample $Y_K$ is $MK$, and $N$ parallel processors are needed. We remark that the implementation of this parallel algorithm assumes that the GPU memory is capable of supporting score estimations for a large batch of data simultaneously. Parallelizing across multiple GPUs introduces additional communication overhead, which may impact efficiency.

**Theoretical guarantee of parallel sampling.** The convergence guarantees for the above parallel sampling is as follows, whose proof can be found in Section 5:

**Theorem 2.** *With the same setting as Theorem 1, it is sufficient to choose*

$$N \gtrsim \frac{d^{1/3} \log^{2/3} T}{\varepsilon^{2/3}}, \qquad MK \gtrsim L \log^2 T, \qquad and \qquad \varepsilon_{\mathsf{score}}^2 \lesssim \varepsilon^2 \log^{-1} T \tag{14}$$

*to achieve* $\mathsf{TV}(p_{\mathsf{data}}, p_{Y_K}) \lesssim \varepsilon$ *for the sampler* (13).

Finally, let us briefly compare our theory with the prior works. This theorem states that the parallel sampler achieves $\varepsilon$-accuracy with respect to total variation distance through using $O(\log^2(Ld/\varepsilon))$ parallel rounds, which is similar to Gupta et al. (2024); Chen et al. (2024a). In addition, we only need $\widetilde{O}(d^{1/3}\varepsilon^{-2/3})$ parallel processors, which improves their results significantly.

## 4 ANALYSIS

This section is devoted to establishing Theorem 1. Before proceeding, we find it helpful to introduce the following notations. Without abuse of notations, we use $X_\tau$ to denote the continuous process as follows:

$$X_\tau \overset{\mathrm{d}}{=} \sqrt{1-\tau} X_0 + \sqrt{\tau} Z, \text{ with } Z \sim \mathcal{N}(0, I), \text{ for } 0 \le \tau \le 1, \tag{15}$$

and the corresponding score function is

$$s_\tau^\star(x) := \nabla \log p_{X_\tau}(x) = -\frac{1}{\tau} \int_{x_0} p_{X_0 \mid X_\tau}(x_0 \mid x)(x - \sqrt{1-\tau} x_0) \mathrm{d}x_0.$$

In addition, we rewrite the sampling process with the continuous index as following:

$$\frac{Y_{\tau_{k,n}}}{\sqrt{1-\tau_{k,n}}} = \frac{Y_k}{\sqrt{1-\tau_{k,0}}} + \frac{s_{\tau_{k,0}}(Y_k)}{2(1-\tau_k)^{3/2}}(\tau_{k,0} - \widehat{\tau}_{k,0}) + \sum_{i=1}^{n-1} \frac{s_{\tau_{k,i}}(Y_{\tau_{k,i}})}{2(1-\tau_{k,i})^{3/2}}(\widehat{\tau}_{k,i-1} - \widehat{\tau}_{k,i})$$
$$+ \frac{s_{\tau_{k,n-1}}(Y_{\tau_{k,n-1}})}{2(1-\tau_{k,n-1})^{3/2}}(\widehat{\tau}_{k,n-1} - \tau_{k,n}),$$

and

$$Y_{k+1} = \sqrt{\frac{1-\tau_{k+1,0}}{1-\tau_{k,N}}} Y_{\tau_{k,N}} + \sqrt{\frac{\tau_{k+1,0} - \tau_{k,N}}{1-\tau_{k,N}}} Z_k,$$

where $Y_0 \sim \mathcal{N}(0, I)$ and $Z_k \overset{\text{i.i.d.}}{\sim} \mathcal{N}(0, I)$. Here, $Y_{\tau_{k,n}}$ corresponds to the $Y_{k,n}$ in the discrete case. In the following, it is enough to consider $T > c_3 L \log T$ for some constant $c_3 > 0$ large enough; otherwise, the desired bound holds trivially.

Before diving into the details of the proof, we provide some core institution of our algorithm and the overview of the proof. Our algorithm is designed based on the following institution. First, by injecting noise into the probability flow ODE (cf. (16)), we transfer the sampling task to one of discretizing the ODE with small error (see Lemma 2). This step contributes to the main improvement of our work over prior works, which provides a simple framework for transferring the sampling task to discretizing the ODE (16) with a small error. Once this reduction is established, we make use of the techniques proposed by Shen & Lee (2019) and Gupta et al. (2024), utilizing randomized midpoints to improve efficiency in estimating the ODE solution (see Lemma 3). The final algorithm is designed based on the discretization form of ODE (16), where estimates of $X_{\tau_{k,i}}$ are iteratively computed using this form and are inserted into the subsequent estimates.

In our algorithm, the main sources of error include discretization error controlled in Lemma 3, estimation error arising from the approximation of $X_{\tau_{k,i}}$, which is bounded in Lemma 4, and the initialization and early stopping errors, which are controlled in Lemma 5 using techniques similar to prior works. The proof can be divided into three steps.

## 4.1 STEP 1: INTRODUCE THE AUXILIARY SEQUENCE

Let $\Phi_{\tau_1 \to \tau_2}(x) := x_{\tau_2} \mid_{x_{\tau_1} = x}$ defined through the following ODE

$$\mathrm{d}\frac{x_\tau}{\sqrt{1-\tau}} = -\frac{s_\tau^\star(x_\tau)}{2(1-\tau)^{3/2}}\mathrm{d}\tau. \tag{16}$$

Then, we have the following result. The detailed proof can be found in Li & Jiao (2024)(Appendix B.2).

**Lemma 2.** *It can be shown that*

$$\Phi_{\tau_1 \to \tau_2}(X_{\tau_1}) \overset{\mathrm{d}}{=} X_{\tau_2}. \tag{17}$$

*Moreover, assume that $\widehat{X}_0 \overset{\mathrm{d}}{=} X_{\tau_{0,0}}$. Then we have for $0 \le k < K$,*

$$\widehat{X}_{k+1} = \sqrt{\frac{1-\tau_{k+1,0}}{1-\tau_{k,N}}} \Phi_{\tau_{k,0} \to \tau_{k,N}}(\widehat{X}_k) + \sqrt{\frac{\tau_{k+1,0} - \tau_{k,N}}{1-\tau_{k,N}}} Z_k \overset{\mathrm{d}}{=} X_{\tau_{k+1,0}}, \tag{18}$$

*where $Z_k \overset{\text{i.i.d.}}{\sim} \mathcal{N}(0, I)$.*

Given the definition of $\widehat{X}_k$, we make the following decomposition:

$$\mathsf{TV}(p_{\mathsf{data}}, p_{Y_K}) \le \mathsf{TV}(p_{\mathsf{data}}, p_{\widehat{X}_K}) + \mathsf{TV}(p_{\widehat{X}_K}, p_{Y_K}), \tag{19a}$$

and

$$\mathsf{TV}^2(p_{\widehat{X}_K}, p_{Y_K}) \le \frac{1}{2}\mathsf{KL}(p_{\widehat{X}_K} \| p_{Y_K}) \le \frac{1}{2}\mathsf{KL}(p_{\widehat{X}_0, \dots, \widehat{X}_K} \| p_{Y_0, \dots, Y_K})$$

$$= \frac{1}{2}\mathsf{KL}\big(p_{\widehat{X}_0}\|p_{Y_0}\big) + \frac{1}{2}\sum_{k=0}^{K-1} \mathbb{E}_{x_k \sim p_{\widehat{X}_k}}\Big[\mathsf{KL}\big(p_{\widehat{X}_{k+1}|\widehat{X}_k}\,(\,\cdot\,|\,x_k)\,\|\,p_{Y_{k+1}|Y_k}\,(\,\cdot\,|\,x_k)\big)\Big],$$

(19b)

where we make use of Pinsker's inequality and the data-processing inequality. In the next step, we will bound the divergence between $p_{\widehat{X}_{k+1}|\widehat{X}_k}$ and $p_{Y_{k+1}|Y_k}$, and the remaining terms can be controlled easily, which is left to the final step.

## 4.2 STEP 2: CONTROL THE DIVERGENCE BETWEEN $p_{\widehat{X}_{k+1}|\widehat{X}_k}$ AND $p_{Y_{k+1}|Y_k}$

Based on the update rule for $Y_{\tau_{k,n}}$, we let

$$\frac{y_{\tau_{k,n}}}{\sqrt{1-\tau_{k,n}}} = \frac{y_{\tau_{k,0}}}{\sqrt{1-\tau_{k,0}}} + \frac{s_{\tau_{k,0}}(y_{\tau_{k,0}})}{2(1-\tau_k)^{3/2}}(\tau_{k,0} - \widehat{\tau}_{k,0}) + \sum_{i=1}^{n-1} \frac{s_{\tau_{k,i}}(y_{\tau_{k,i}})}{2(1-\tau_{k,i})^{3/2}}(\widehat{\tau}_{k,i-1} - \widehat{\tau}_{k,i})$$
$$+ \frac{s_{\tau_{k,n-1}}(y_{\tau_{k,n-1}})}{2(1-\tau_{k,n-1})^{3/2}}(\widehat{\tau}_{k,n-1} - \tau_{k,n}),$$

for $n = 1, \ldots, N$. Then notice that $\widehat{X}_{k+1}|\widehat{X}_k$ and $Y_{k+1}|Y_k$ are both normal distributions with the same variance $\frac{\tau_{k+1,0}-\tau_{k,N}}{1-\tau_{k,N}}$ and different means $\sqrt{\frac{1-\tau_{k+1,0}}{1-\tau_{k,N}}}\Phi_{\tau_{k,0}\to\tau_{k,N}}(\widehat{X}_k)$ and $\sqrt{\frac{1-\tau_{k+1,0}}{1-\tau_{k,N}}}y_{\tau_{k,N}}$, respectively. This tells us that

$$\mathsf{KL}\big(p_{\widehat{X}_{k+1}|\widehat{X}_k}\,(\,\cdot\,|\,x_k)\,\|\,p_{Y_{k+1}|Y_k}\,(\,\cdot\,|\,x_k)\big) = \frac{1-\tau_{k+1,0}}{2(\tau_{k+1,0} - \tau_{k,N})}\|y_{\tau_{k,N}} - x_{\tau_{k,N}}\|_2^2, \quad (20)$$

when we set $x_{\tau_{k,0}}, y_{\tau_{k,0}} = x_k$, and $x_{\tau_{k,N}}$ is defined as $x_{\tau_{k,N}} = \Phi_{\tau_{k,0}\to\tau_{k,n}}(x_k)$.

Towards controlling the difference between $y_{\tau_{k,N}}$ and $x_{\tau_{k,N}}$, let's consider the ODE trajectory beginning from $x_{\tau_{k,0}} = x_k$, and define

$$\xi_{k,n}(x_k) := \frac{x_{\tau_{k,n}}}{\sqrt{1-\tau_{k,n}}} - \frac{x_{\tau_{k,0}}}{\sqrt{1-\tau_{k,0}}} - \frac{s_{\tau_{k,0}}(x_{\tau_{k,0}})}{2(1-\tau_{k,0})^{3/2}}(\tau_{k,0} - \widehat{\tau}_{k,0})$$
$$- \sum_{i=1}^{n-1} \frac{s_{\tau_{k,i}}(x_{\tau_{k,i}})}{2(1-\tau_{k,i})^{3/2}}(\widehat{\tau}_{k,i-1} - \widehat{\tau}_{k,i}) - \frac{s_{\tau_{k,n-1}}(x_{\tau_{k,n-1}})}{2(1-\tau_{k,n-1})^{3/2}}(\widehat{\tau}_{k,n-1} - \tau_{k,n}),$$

(21)

which denotes the discretization and estimation error of the ODE process (16).

Given an initialization point $x_{\tau_{k,0}} = x_k$, the term $\theta(\tau; x_k) := \frac{s_\tau^\star(x_\tau)}{2(1-\tau)^{3/2}}$ depends only on $\tau$. Solving (16) is equivalent to calculating the integral of $\theta(\tau; x_k)$. Consequently, it is natural for the algorithm to approximate the integral by discretizing $\tau$ and estimating $\theta(\tau; x_k)$ at some discrete time points. The following lemma controls the estimation error, and the underlying intuition lies in controlling the error between the true value and its estimation obtained through this discretization as defined in (21). The detailed proof can be found in Li & Jiao (2024)(Appendix A.1). Similar errors are also analyzed in Gupta et al. (2024) (Appendix A.1) under an exponential integrator formulation. Our derivation adapts these ideas to the context of the probability flow ODE discretization in this work.

**Lemma 3.** *For any $k$ and $n$, it can be seen that with probability at least $1 - T^{-100}$,*

$$\mathbb{E}_{x_k \sim p_{\widehat{X}_k}}\big[\|\xi_{k,n}(x_k)\|_2^2\big] \lesssim \sum_{i=1}^{n} \frac{L^2 d \log^4 T}{(1-\widehat{\tau}_{k,i-1})^2 T^3}(\widehat{\tau}_{k,i-1} - \widehat{\tau}_{k,i})$$
$$+ \frac{N \log^2 T}{T^2}\sum_{i=0}^{n-1}\widehat{\tau}_{k,i}(1-\widehat{\tau}_{k,i})^{-1}\varepsilon_{T-\frac{kN}{2}-i+1}^2, \quad (22)$$

*where $\varepsilon_{T-\frac{kN}{2}-i+1}^2$ is defined in (6).*

With the above relation, we can bound the divergence as following. The detailed proof can be found in Li & Jiao (2024)(Appendix A.2).

**Lemma 4.** *According to Lemma 3, it can be shown that*

$$
\mathbb{E}_{x_k \sim p_{\widehat{X}_k}} \left[ \mathsf{KL} \left( p_{\widehat{X}_{k+1}|\widehat{X}_k} \left( \cdot \mid x_k \right) \| p_{Y_{k+1}|Y_k} \left( \cdot \mid x_k \right) \right) \right] \lesssim \frac{L^2 d \log^4 T}{T^3} + \frac{\log T}{T} \sum_{i=0}^{N-1} \varepsilon^2_{T - \frac{kN}{2} - i + 1},
$$

(23)

*where $\varepsilon^2_{T - \frac{kN}{2} - i + 1}$ is defined in (6).*

### 4.3 STEP 3: PUTTING EVERYTHING TOGETHER

The remaining terms in (19) can be bounded through the following lemma. The detailed proof can be found in Li & Jiao (2024)(Appendix B.3).

**Lemma 5.** *Under our choice of learning schedule (9), we have*

$$
\mathsf{TV}\big(p_{\mathsf{data}}, p_{\widehat{X}_K}\big) \leq \frac{1}{T^{10}} \quad and \quad \mathsf{KL}\big(p_{\widehat{X}_0} \| p_{Y_0}\big) \leq \frac{1}{T^{10}}.
$$

(24)

Inserting (24) and (23) into (19) leads to

$$
\mathsf{TV}\left(p_{\mathsf{data}}, p_{Y_K}\right) \lesssim \frac{1}{T^{10}} + \sqrt{\frac{1}{T^{10}} + \frac{L^3 d \log^5 T}{T^3} + \varepsilon^2_{\mathsf{score}} \log T} \lesssim \frac{L^{3/2} d^{1/2} \log^{5/2} T}{T^{3/2}} + \varepsilon_{\mathsf{score}} \log^{1/2} T,
$$

and we conclude the proof here.

## 5 ANALYSIS FOR PARALLELIZATION (THEOREM 2)

By comparing the update rules for $Y_{k,n}$ and $Y_{m,k,n}$, it is natural to control the difference of the following two sequences:

$$
\frac{y_{\tau_{k,n}}}{\sqrt{1 - \tau_{k,n}}} = \frac{y_{\tau_{k,0}}}{\sqrt{1 - \tau_{k,0}}} + \frac{s_{\tau_{k,0}}(y_{\tau_{k,0}})}{2(1 - \tau_{k,0})^{3/2}}(\tau_{k,0} - \widehat{\tau}_{k,0}) + \sum_{i=1}^{n-1} \frac{s_{\tau_{k,i}}(y_{\tau_{k,i}})}{2(1 - \tau_{k,i})^{3/2}}(\widehat{\tau}_{k,i-1} - \widehat{\tau}_{k,i})
$$
$$
+ \frac{s_{\tau_{k,n-1}}(y_{\tau_{k,n-1}})}{2(1 - \tau_{k,n-1})^{3/2}}(\widehat{\tau}_{k,n-1} - \tau_{k,n}),
$$

and

$$
\frac{y_{m,\tau_{k,n}}}{\sqrt{1 - \tau_{k,n}}} = \frac{y_{\tau_{k,0}}}{\sqrt{1 - \tau_{k,0}}} + \frac{s_{\tau_{k,0}}(y_{\tau_{k,0}})}{2(1 - \tau_{k,0})^{3/2}}(\tau_{k,0} - \widehat{\tau}_{k,0}) + \sum_{i=1}^{n-1} \frac{s_{\tau_{k,i}}(y_{m-1,\tau_{k,i}})}{2(1 - \tau_{k,i})^{3/2}}(\widehat{\tau}_{k,i-1} - \widehat{\tau}_{k,i})
$$
$$
+ \frac{s_{\tau_{k,n-1}}(y_{m-1,\tau_{k,n-1}})}{2(1 - \tau_{k,n-1})^{3/2}}(\widehat{\tau}_{k,n-1} - \tau_{k,n}).
$$

The above update rules, together with the Lipschitz condition of score estimates, lead to

$$
\frac{\left\| y_{m,\tau_{k,n}} - y_{\tau_{k,n}} \right\|_2}{\sqrt{1 - \tau_{k,n}}} \lesssim \frac{L \log T}{T} \sum_{i=1}^{n-1} \frac{\left\| y_{m-1,\tau_{k,i}} - y_{\tau_{k,i}} \right\|_2}{\sqrt{1 - \tau_{k,i}}}.
$$

Applying the above relation recursively gives

$$
\max_n \frac{\left\| y_{M,\tau_{k,n}} - y_{\tau_{k,n}} \right\|_2}{\sqrt{1 - \tau_{k,n}}} \leq \left( \frac{NL \log T}{T} \right)^M \max_n \frac{\left\| y_{0,\tau_{k,n}} - y_{\tau_{k,n}} \right\|_2}{\sqrt{1 - \tau_{k,n}}} \leq \frac{1}{\mathsf{poly}(T)},
$$

provided that $M \gtrsim \log T$ and $T \gtrsim NL \log T$. Thus as long as $N \gtrsim \frac{d^{1/3} \log^{2/3} T}{\varepsilon^{2/3}}$, which guarantees that $T \gtrsim \frac{L d^{1/3} \log^{5/3} T}{\varepsilon^{2/3}}$, we can get the desired result immediately through just inserting the above error boound into Lemma 4. We omit the details here due to the similarity.

## 6 DISCUSSION

In this paper, we establish a faster convergence rate for diffusion probabilistic models, which scales as $Ld^{1/3}\varepsilon^{-2/3}$, assuming smooth score functions. Our theory achieves at least $L^{2/3}d^{1/12}\varepsilon^{-1/3}$ improvement over prior works (Gupta et al., 2024; Chen et al., 2024b), and thereby aligns more closely with practical requirements. Recent researches (Benton et al., 2023; Li et al., 2024c) have shown an $O(d\varepsilon^{-1})$ dependence for diffusion models without the assumption of smoothness. However, it remains unclear whether any algorithm can achieve a sub-linear dependence on $d$ in the absence of smoothness. We leave this as an open question for future research. Moreover, the benefit of the randomized design in our work relies heavily on the deterministic nature of ODE process. It is still unclear for us how to adapt this approach to deal with the inherent stochasticity in SDEs while maintaining similar improvements. This will also be left as future work. In addition, it may be feasible to apply your analysis framework to improve the bound for other variants of samplers such as the Langevin algorithm. Furthermore, estimating the Lipschitz constant in real-world cases would be highly beneficial, potentially broadening the applicability of our results.

### ACKNOWLEDGMENTS

Gen Li is supported in part by the Chinese University of Hong Kong Direct Grant for Research and the Hong Kong Research Grants Council ECS 2191363.

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
