# OpenReview forum: "Improved Convergence Rate for Diffusion Probabilistic Models"
_ICLR.cc/2025/Conference — ICLR 2025 Spotlight_

### Official Review · Reviewer_FYHg · 2024-11-02

**Soundness:** 4
**Presentation:** 2
**Contribution:** 4
**Rating:** 8
**Confidence:** 4

**Summary:**

This paper considers the well-studied question of sampling from a smooth distribution $p$ given approximate access to its score functions $\nabla \ln p_t$, where $p_t$ is $p$ convolved with some level of noise.

Recent work has shown that this can be done for essentially arbitrary smooth distributions, with iteration complexity $L^a d^b/\epsilon^{c}$ for progressively smaller constants $a,b,c$. Here $L$ is the Lipschitz constant for the score functions and score estimates. Recent works of Chen et al. and Gupta et al. culminated in bounds of $(a,b,c) = (2/3, 5/12, 1)$. The present work gives a clever analysis that pushes this down to $(1, 1/3, 2/3)$.

While not explicitly written as such, the sampler the algorithms propose can be thought of as a predictor-corrector scheme: it alternates between reverse diffusion steps (implemented via a slight variant of the randomized midpoint method) and steps in which additional noise is added to the trajectory. The corrector steps allow one to essentially convert Wasserstein closeness between the algorithm and the true (continuous) reverse diffusion into KL closeness, so that the KL between the algorithm and the true distribution is small by chain rule for KL.

**Strengths:**

This result was quite surprising to me. It is a big open question to achieve $d^{1/3}$ dependence for log-concave sampling in KL, and I had expected that the result in the present paper would not be achieved until that milestone had been reached.

The key insight, which is somewhat buried in the notation, is the following. The prior works of Chen et al. and Gupta et al. similarly adopted a predictor-correct approach, but their corrector was based on running underdamped Langevin dynamics to keep the sampler stationary at a fixed noise level. But this required a discretization analysis for underdamped Langevin in KL; while it is conjectured that one can achieve $d^{1/3}$ dependence for this (and indeed, this is the best known rate one can achieve for Wasserstein), the best known analyses are currently stuck at $d^{1/2}$.

The authors of the present work sidestep this with an ingenious idea: instead of running underdamped Langevin at a fixed noise level, they simply add Gaussian noise in the corrector step and rescale. It is much more straightforward to see that this converts Wasserstein closeness to KL closeness. Instead of preserving the noise level however, this has the effect of increasing the noise slightly. But if the rescaling is chosen appropriately, it still remains on the trajectory of the reverse diffusion! By balancing appropriately between the progress the predictor step makes in reducing the noise level and the progress the corrector step undoes by raising the noise level, the authors are able to circumvent the issues from prior work and achieve the desired $d^{1/3}$ scaling. This idea of using a corrector step which raises the noise level while staying on the trajectory of the true reverse diffusion is simple and very nice.

**Weaknesses:**

My main complaint is that the writing is very dense and it was very hard from a quick glance to extract the key idea outlined above. If this work is to be accepted, the authors should make sure to include a technical overview section where they explain the intuition for how they are able to go beyond the prior works of Chen et al. and Gupta et al.

There might be some complaints about the fact that they need to assume Lipschitzness of the score and the score estimate, but note that no sublinear-in-$d$ bounds are known for diffusions under less stringent assumptions, so I wouldn't count this as a real downside.

**Questions:**

I'm not sure I followed why one needs to modify randomized midpoint for the "predictor" part of the sampler, i.e. in the computation of the $Y_{k,n}$'s. Why couldn't one just run the predictor analysis in Gupta et al. off the shelf to get an analogue of Lemma 3?

---

> ### Author Response · Authors · 2024-11-18
> **rebuttal**
>
> Thanks a lot for your detailed and positive comments on our paper! We sincerely appreciate the time you took to review our work. Below, we address your points in detail:
>
> 1. Technical overview to explain the intuition
>
> Thanks a lot for your explicit summary. The main improvement of our work over prior works is based on Lemma 2, which provides a simple framework for transferring the sampling task to discretizing the ODE (16) with a small $\ell_2$ error. Once this reduction is established, we leverage the randomized midpoint method, as in prior works, to achieve a more efficient discretization of the ODE and attain the desired bounds.
>
> We will incorporate a detailed technical overview in the revised manuscript, to explain the intuition following your summary.
>
> 2. Discussion of non-Lipschitz setting
>
> We will add a remark to explain the difficulty in removing the smoothness condition as follows: The extension to non-Lipschitz settings is significantly more challenging due to the critical role the smoothness assumption plays in our analysis. Previous analysis on diffusion models focus primarily on stepwise error propagation, analyzing how errors at one step affect the next. In contrast, our randomized design requires tracking error propagation across multiple steps simultaneously, which necessitates uniform control over these errors. Without smoothness, only a high-probability bound that depends on the dimension $d$ can be established, which is  hard to use compared to the uniform bound and may result in worse bound since it scales with $d$.
>
> 3. Modification for Lemma 3 compared to Gupta et al.
>
> We are sorry for missing this discussion regarding Lemma 3. While similar bounds are indeed presented in Gupta et al. (Appendix A.1), they are derived under an exponential integrator formulation. In our work, we derive Lemma 3 within the context of our framework and notations, tailored specifically to our method.
>
> We will include the following remark in the revised manuscript: Similar bounds as Lemma 3 are also presented in Gupta et al. (Appendix A.1) under an exponential integrator formulation. Our derivation adapts these ideas to the context of the probability flow ODE discretization in this work.

---

> > ### Comment · Reviewer_FYHg · 2024-11-23
> > **Thanks for the response!**
> >
> > I remain very positive about this work and will keep my score, thanks for answering my questions!

---

> > > ### Author Response · Authors · 2024-11-29
> > >
> > > Thanks again for your efforts in reviewing our paper and for your positive feedback!

---

### Official Review · Reviewer_Twhm · 2024-11-02

**Soundness:** 3
**Presentation:** 2
**Contribution:** 4
**Rating:** 8
**Confidence:** 4

**Summary:**

This paper studies the problem of sampling from a diffusion model, and provides an algorithm with the best known dimension dependence of $d^{1/3}$, an improvement over $d^{5/12}$ of previous work. It also improves the dependence on the score estimation error $\epsilon$, from $1/\epsilon$ to $\epsilon^{-2/3}$. The algorithm uses the randomized midpoint method along with corrector steps, a strategy first proposed for diffusion by Gupta et. al. (2024). The authors show that a new analysis of this algorithm (with some modifications) achieves the improved iteration complexity bounds.

**Strengths:**

Studies an important and well-studied problem and develops new techniques that allow the authors to show the best known iteration complexity. Perhaps surprisingly, as of this work, we have faster algorithms for sampling from diffusion models than for log-concave sampling.

**Weaknesses:**

While I appreciate the brevity of the paper, I feel that the core intuition could be explained better for easier digestion.

Overall though, I don't have many complaints.

**Questions:**

- Is it possible to translate these techniques to obtain improved bounds for log-concave sampling in TV? Please update the paper to discuss any challenges in doing so.

- Can you condense the core intuition and explain it in a proof overview? Please add a proof overview to the paper that explains the key insight relative to prior work in a condensed form.

- Does this work have any implications for the setting where the score is not assumed to be Lipschitz? Please add a discussion of the challenges in translating these techniques to the non-Lipschitz setting, or what bound you could hope to get.

---

> ### Author Response · Authors · 2024-11-18
> **rebuttal**
>
> Thanks a lot for your positive comments! We sincerely appreciate your time for reviewing our work. Below, we address your questions and suggestions in detail:
>
> 1. Core intuition and proof overview
>
> The overarching framework of this paper is as follows. First, by injecting noise into the probability flow ODE (16), we transfer the sampling task to one of discretizing the ODE (16) with small $\ell_2$ error (see Lemma 2). Then we make use of the techniques proposed by Shen & Lee, (2019) and Gupta et al. (2024), utilizing randomized midpoints to improve efficiency in estimating the ODE solution (see Lemma 3). The final algorithm is designed based on the discretization form of ODE (16), where estimates of $x_{\tau_{k, i}}$ are iteratively computed using this form and are inserted into the subsequent estimates. In addition, the main sources of error include: discretization error controlled in Lemma 3, estimation error arising from the approximation of $x_{\tau_{k, i}}$, which is bounded in Lemma 4, and the initialization and early stopping errors, which are controlled in Lemma 5 using techniques similar to prior works.
>
> To enhance clarity, we will include a detailed explanation for the core intuition and a proof overview in the revised manuscript.
>
> 2. Extension to log-concave sampling
>
> Thanks a lot for your suggestion regarding log-concave sampling. As our result relies only on the smoothness condition, the total variation (TV) bound $d^{1/3}\epsilon^{-2/3}$ in Theorem 1 holds true directly for the log-concave case. Hense, our TV bound improves upon prior works, such as Theorem 1.3 in Gupta et al. (2024), by at least $d^{1/12}$, without requiring the stongly-log-concave condition.
>
> We will add a detailed discussion on this extension in the revised manuscript.
>
> 3. Discussion on the non-Lipschitz setting
>
> Extending our techniques to the more general case is more challenging due to the absence of the smoothness assumption. In most previous analyses of diffusion models, the primary focus is on the stepwise influence of error: how errors propagate from one step to the next. However, obtaining the benefit of the random design in our method requires tracking error propagation across multiple steps simultaneously. This multi-step dependency critically depends on the smoothness condition, which allows uniform control over error propagation. Without smoothness, the analysis can only guarantee some high probability bound that depends on the dimension $d$. This bound is hard to use compared to the uniform bound, and may lead to a worse bound since it scales with $d$.
>
> We will add a remark in the manuscript to clarify the difficulties of extending our approach to the non-Lipschitz setting and the limitations of current techniques in this regard.

---

> > ### Comment · Reviewer_Twhm · 2024-11-21
> >
> > I don't think I understand your comment about the log-concave case? I agree that diffusion can handle non-log-concave distributions, but what I am asking here is whether your techniques translate to an improved bound for standard Langevin for a log-concave distribution in total variation. Note that Gupta et al showed a bound of $ O(d^{5/12})$ for this problem in their paper, using techniques similar to the ones used in the diffusion results. I am wondering if your new analysis can be applied to improve the bound for this problem to $O(d^{1/3})$.

---

> > > ### Author Response · Authors · 2024-11-24
> > >
> > > We are sorry for the misunderstanding. Our results specifically focus on diffusion models rather than the Langevin algorithm. We believe it is feasible to derive an improved discretization error bound for the Langevin algorithm based on our observations for diffusion models due to their similarity. However, this would require additional efforts and a separate analysis. Moreover, this direction falls out of the scope of our current study, which aims to provide an improved bound for diffusion models. Thus we leave it as future work.

---

> > > > ### Comment · Reviewer_Twhm · 2024-11-27
> > > >
> > > > OK, I maintain my score. Thanks.

---

### Official Review · Reviewer_yjdB · 2024-11-04

**Soundness:** 3
**Presentation:** 2
**Contribution:** 3
**Rating:** 8
**Confidence:** 4

**Summary:**

This paper studies the diffusion models and proposes sampling method based on randomized midpoint for probability flow ODE.  Their method has an improved sequential iteration complexity as $Ld^{1/3}/\epsilon^{-2/3}$ and similar parallel iteration complexity $\log^2(d/\epsilon)$, with measurement as total variance.

**Strengths:**

- The paper studies the complexity of the inference process of diffusion models, which is a problem of significane interest and practical importance in the related fields.

- the sequential itertaion complexity is improved from $L^{5/3}d^{5/12}\epsilon^{-1}$ to $Ld^{1/3}\epsilon^{-2/3}$.

**Weaknesses:**

- This main drawback is the additional smoothness assumption on the score function and its estimation.

- The algorithm and the proof presented in Lemma 3 are difficult to follow. It may be helpful to provide more detailed explanations or include additional diagrams for clarity.

- Your method looks like N=poly(T)-stage Runge-Kutta method with random Runger-Kutta matrix. Then $\varepsilon^{-1/N}$ iteration complexity is expected although such method needs higher order smooth for ode case [1]. Calling it the midpoint method seems somewhat misleading.

[1]Convergence analysis of probability flow ODE for score-based generative models, Huang, Daniel Zhengyu, Jiaoyang Huang, and Zhengjiang Lin, 2024.

**Questions:**

- I am curious about the results when your method is applied to the implementation of diffusion models using SDEs

---

> ### Author Response · Authors · 2024-11-18
> **rebuttal**
>
> We are grateful for your time for reviewing our paper. Below, we provide point-by-point responses to your comments and questions:
>
> 1. Smoothness condition
>
> We agree that the requirement for a smoothness condition on the score function and its estimation is a limitation of our theoretical framework. However, we want to argue that our result provides improved convergence compared with prior results under general conditions in many cases, such as $L < 3381$ for $\epsilon = 1$ and $L < 72845$ for $\epsilon = 0.1$, as discussed in Section 1.2 and 3.2. Addressing the challenge of achieving sublinear dependency on $d$ without imposing the smoothness assumption needs much more extra effort, and will be left as our future research.
>
> 2. More explanations for algorithm and proof in Lemma 3
>
> The core of the algorithm, as described in Lemma 2, involves discretizing the ODE (16) to generate $\widehat{X}_{k+1}$ from $\widehat{X}_k$. Given an initialization point $x\_{\tau\_{k,0}}=x_k$, the term $\theta(\tau; x_k) := \frac{s\_{\tau}^{\star}(x\_{\tau})}{2(1-\tau)^{3/2}}$ depends only on $\tau$. Solving (16) is equivalent to calculating the integral of $\theta(\tau; x_k)$. Consequently, it is natural for the algorithm to approximate the integral by discretizing $\tau$ and estimating $\theta(\tau; x_k)$ at some discrete time points. The intuition behind Lemma 3 lies in controlling the error between the true value $x\_{\tau\_{k, n}}$ and its estimation obtained through this discretization as defined in (21).
>
> As for the proof of Lemma 3, we decompose the error into four terms, each of which is analyzed separately. The first two terms correspond to errors at the intial and end points, the second term contains the discretization error for all middle points, and the last term includes estimation errors. The second term is the main term and its analysis leverages the randomized design.
>
> We will add a detailed explanation for Lemma 3 in the revised manuscript.
>
> 3. Connection with random RK method
>
> Thanks a lot for pointing out the connection with random designed RK method. We will cite and compare with [1] appropriately. Here our method is termed ''randomized midpoint'' following the literature: it aligns with its initial introduction in [2] for log-concave sampling, and subsequent adaptation for diffusion models by [3].
>
> 4. Extension to SDE-type sampler
>
> This is a quite interesting direction. The benefit of the randomized design in our work relies heavily on the deterministic nature of ODE process (16). It is still unclear for us how to adapt this approach to deal with the inherent stochasticity in SDEs while maintaining similar improvements. This will be left as our future work.
>
> [1] Huang, Daniel Zhengyu, Jiaoyang Huang, and Zhengjiang Lin, Convergence analysis of probability flow ODE for score-based generative models, 2024.
>
> [2] Ruoqi Shen and Yin Tat Lee. The randomized midpoint method for log-concave sampling, Advances in Neural Information Processing Systems, 2019.
>
> [3] Shivam Gupta, Linda Cai, and Sitan Chen. Faster diffusion-based sampling with randomized midpoints: Sequential and parallel, arXiv:2406.00924, 2024.

---

> > ### Comment · Reviewer_yjdB · 2024-11-25
> >
> > Thank you for your response. While [1] highlights infinite Lipschitz behavior near zero, I still consider the smoothness condition a key limitation. Overall, I think it is a nice paper with multi-step method, and I will increase the score.
> >
> > [1] Eliminating Lipschitz Singularities in Diffusion Models, Yang et al., 2024

---

> > > ### Author Response · Authors · 2024-11-26
> > >
> > > Thanks for your acknowledgment. In the final version, we will provide a more detailed discussion of the smoothness condition.

---

### Official Review · Reviewer_EyNz · 2024-11-04

**Soundness:** 3
**Presentation:** 3
**Contribution:** 3
**Rating:** 6
**Confidence:** 3

**Summary:**

The paper focuses on enhancing the theoretical convergence rate of diffusion models, which are widely used for high-quality data generation in various machine learning applications. Traditional analyses show a significant gap between theoretical complexity and the number of steps needed in practice for good performance. This work introduces a faster convergence rate of $O\left(d^{1 / 3} \epsilon^{-2 / 3}\right)$, an improvement over previous bounds, achieved by adapting a randomized midpoint method. The authors' analysis allows for smooth, $\epsilon$-accurate score estimates without assuming restrictive properties like log-concavity in the data distribution. Additionally, they propose a parallel sampling technique that minimizes the number of parallel rounds required, further increasing efficiency.

**Strengths:**

1. Theorem 1 introduces a new convergence rate for diffusion models, specifically $O\left(d^{1 / 3} \epsilon^{-2 / 3}\right)$, which is faster than previous results (e.g., $O\left(d^{5 / 12} \epsilon^{-1}\right)$ by Gupta et al., 2024). This improvement relies on an adaptation of the randomized midpoint method from log-concave sampling and brings theoretical guarantees closer to practical performance.
2. Theorem 2 paper presents an efficient parallel sampling method that reduces the number of parallel rounds needed while maintaining the improved convergence rate.

**Weaknesses:**

1. Theorem 2 describes an efficient parallel sampling method with fewer parallel rounds, but the practical implementation details are limited. like How does memory consumption scale with $d$ in parallel sampling? Are there specific cases or datasets where the parallel method might fail to produce substantial speedups due to communication overhead between processors? If so, what alternative strategies or adjustments would the authors suggest?
2. Are there specific types of data distributions (e.g., multimodal, heavy-tailed) that the authors believe may not meet the framework’s assumptions?

**Questions:**

See weaknesses part above.

---

> ### Author Response · Authors · 2024-11-18
> **rebuttal**
>
> We sincerely appreciate your time for reviewing our paper. Below, we address your comments point by point:
> 1. Practical implementation details.
>
> In practice, the neural network evaluations of score functions are typically conducted on batches (denoted as $B$) of data. This enables our algorithm to be implemented with at least $B$ parallelized processes without incurring communication overhead. As you said, the primary bottleneck in this setup is GPU memory capacity, which scales near linearly in $B = d^{1/3}\epsilon^{-2/3}$ in our Theorem 2. Moreover, we believe that the communication costs among multiple GPUs are negligible compared to the computation costs, as it is common practice to train and infer large models across many GPUs.
>
> For example, using a single NVIDIA V100 GPU (32GB), it is feasible to set a batch size of over $50$ for latent diffusion model sampling on datasets such as FFHQ and LSUN (church/bedroom) [1]. This implies that the score estimates $s(x_t, t)$ for more than $50$ different $(x_t, t)$ pairs can be computed simultaneously, resulting in an approximately $50$-fold acceleration. This performance is sufficient to handle lower-accuracy scenarios where $\epsilon > 0.3$, ensuring that $d^{1/3}\epsilon^{-2/3} < B = 50$ for $d < 12288$.
>
> To address the limitations of this parallel algorithm, we will include the following remark in the revised manuscript: "The implementation of this parallel algorithm assumes that the GPU memory is capable of supporting score estimations for a large batch of data simultaneously. Parallelizing across multiple GPUs introduces additional communication overhead, which may impact efficiency."
>
> [1] Rombach, Robin, et al. High-resolution image synthesis with latent diffusion models. Proceedings of the IEEE/CVF conference on computer vision and pattern recognition. 2022.
>
>
> 2. Data assumption
>
> Our framework imposes two assumptions on data: bounded second order moment and smoothness, which we believe encompass many practical applications. However, we acknowledge that exceptions exist:
>
> The assumption of bounded second-order moments can accommodate many heavy-tailed distributions as long as their density decays no slower than $1/x^3$. This assumption fails only for distributions with density scales at the order of $1/x^c$ for $1<c<3$.
>
> The smoothness assumption excludes non-smooth (e.g., discrete) distributions. It remains valid for certain multimodal distributions. For instance, if the data are generated from a Gaussian Mixture Model (GMM), $X_0 \sim \sum_i \pi_i\mathcal{N}(\mu_i, 1)$, then the score function remains smooth.
>
> We will include the following remark in the revised manuscript to explain these assumptions: ''Our assumptions of bounded second-order moments and smoothness hold for many real-world distributions. However, there exist exceptions including extremely heavy-tailed or non-smooth distributions. For example, densities decaying slower than $1/x^3$ or purely discrete data may not align with our theoretical framework.''

---

> > ### Author Response · Authors · 2024-11-29
> >
> > Thanks again for your efforts in reviewing our paper and for your helpful comments! We have carefully considered your questions and addressed them in our response. The discussion phase is due on December 2nd, and we would like to know whether our response has appropriately addressed your questions and concerns about our paper. If we have addressed your concerns, we would appreciate it if you consider increasing your score for our paper. Please let us know if you have further comments or concerns about our paper. Thank you!

---

### Meta-Review · Area_Chair_SfZ4 · 2024-12-15

**Metareview:**

This paper studies the sampling from a diffusion model, providing an algorithm with iteration complexity of order $d^{1/3}\varepsilon^{-2/3}$, which improves upon the order $d^{5/12}\varepsilon^{-1}$ of previous work. The improvement in the dimension dependence is particularly impressive given that it is an important open question to achieve dependence $d^{1/3}$ for log-concave sampling in KL.

The reviewers appreciated the strong results and the neat idea at their basis (instead of running underdamped Langevin at a fixed noise level, add Gaussian noise in the corrector step and rescale). There was clear enthusiasm in the reviews and I agree with the overall positive evaluation. This is a strong paper that will be appreciated at ICLR.

**Additional Comments On Reviewer Discussion:**

No important issues were raised during the reviews, but mostly comments about clarity.

---

### Decision · Program_Chairs · 2025-01-22

Accept (Spotlight)